# A LAMP sequencing approach for high-throughput co-detection of SARS-CoV-2 and influenza virus in human saliva

Robert Warneford-Thomson[1,2,3], Parisha P Shah[2,3,4], Patrick Lundgren[5], Jonathan Lerner[3], Jason Morgan[6], Antonio Davila[6,7], Benjamin S Abella[6], Kenneth Zaret[2,3], Jonathan Schug[8], Rajan Jain[2,3,4], Christoph A Thaiss[5], Roberto Bonasio[2,3]*

[1]Graduate Group in Biochemistry and Biophysics, University of Pennsylvania Perelman School of Medicine, Philadelphia, United States; [2]Epigenetics Institute, University of Pennsylvania Perelman School of Medicine, Philadelphia, United States; [3]Department of Cell and Developmental Biology, University of Pennsylvania Perelman School of Medicine, Philadelphia, United States; [4]Department of Medicine, University of Pennsylvania Perelman School of Medicine, Philadelphia, United States; [5]Department of Microbiology, University of Pennsylvania Perelman School of Medicine, Philadelphia, United States; [6]Department of Emergency Medicine and Penn Acute Research Collaboration, University of Pennsylvania Perelman School of Medicine, Philadelphia, United States; [7]University of Pennsylvania School of Nursing, Philadelphia, United States; [8]Next-Generation Sequencing Core, Department of Genetics, University of Pennsylvania Perelman School of Medicine, Philadelphia, United States

*For correspondence:
roberto@bonasiolab.org

**Abstract** The COVID-19 pandemic has created an urgent need for rapid, effective, and low-cost SARS-CoV-2 diagnostic testing. Here, we describe COV-ID, an approach that combines RT-LAMP with deep sequencing to detect SARS-CoV-2 in unprocessed human saliva with a low limit of detection (5–10 virions). Based on a multi-dimensional barcoding strategy, COV-ID can be used to test thousands of samples overnight in a single sequencing run with limited labor and laboratory equipment. The sequencing-based readout allows COV-ID to detect multiple amplicons simultaneously, including key controls such as host transcripts and artificial spike-ins, as well as multiple pathogens. Here, we demonstrate this flexibility by simultaneous detection of 4 amplicons in contrived saliva samples: SARS-CoV-2, influenza A, human *STATHERIN*, and an artificial SARS calibration standard. The approach was validated on clinical saliva samples, where it showed excellent agreement with RT-qPCR. COV-ID can also be performed directly on saliva absorbed on filter paper, simplifying collection logistics and sample handling.

## Editor's evaluation

Surveillance screening can help us estimate the prevalence of SARS-CoV-2 infection and co-infection with other respiratory pathogens. This work offers a high-throughput and cost-effective method to do such surveillance based on RT-LAMP combined with deep sequencing. This method can be applied to clinical samples for an accurate reading of the fraction of infections where the SARS-CoV-2 titer is moderate or high.

## Introduction

In two years, the COVID-19 pandemic has swept across the world, leading to more than 490 million infections and over 6.1 million deaths worldwide (as of April 2022). In many countries, non-pharmaceutical interventions, such as school closures and national lockdowns, have proven to be effective, but could not be sustained due to economic and social impact (*Haug et al., 2020*; *Tian et al., 2020*). Regularly performed population-level diagnostic testing is an attractive solution (*Taipale et al., 2020*), particularly as asymptomatic individuals are implicated in rapid disease transmission, with a strong overdispersion in secondary transmission (*Endo et al., 2020*). Sustained population-level testing can be successful in isolating asymptomatic individuals and decreasing transmission (*Holt, 2020*; *Larremore et al., 2020*); however, considerable barriers exist to the adoption of such massive testing strategies. Two such barriers are cost and supply constraints for commercial testing reagents, both of which make it impractical to test large numbers of asymptomatic individuals on a recurrent basis. A third major barrier is the lack of 'user-friendly' protocols that can be rapidly adopted by public and private organizations to establish high-throughput surveillance screening. In addition, while COVID-19 testing of symptomatic individuals might be effective during the summer season, when other respiratory infections are rare, new strategies are needed to facilitate rapid differential diagnosis between SARS-CoV-2 and other respiratory viruses in winter. Although the wide availability of self-administered lateral flow tests has greatly facilitated the identification and isolation of active infections, these tests lack the sensitivity of nucleic acid detection (*Brümmer et al., 2021*).

Recent adaptations of reverse transcription and polymerase chain reaction (RT-PCR) to amplify viral sequence and perform next-generation DNA sequencing have opened promising new avenues for massively parallel SARS-CoV-2 detection. In general, sequencing-based protocols use libraries of amplification primers to tag reads originating from each individual patient sample with a unique index that can be identified and deconvoluted after sequencing, thus allowing pooling of tens of thousands of samples in a single assay. Several methods, including SARSeq, SPAR-Seq, Swab-seq, COVseq, and INSIGHT directly amplify the viral RNA by RT-PCR and simultaneously introduce barcodes (*Simonetti et al., 2021*; *Bloom et al., 2021*; *Yelagandula et al., 2021*; *Aynaud et al., 2021*; *Wu et al., 2021*; *de Mello Malta et al., 2021*). While effective, these methods rely on individual PCR amplification of each patient sample, thus requiring a large number of thermal cyclers for massive scale-up. An alternative approach, ApharSeq, addresses this bottleneck by annealing barcoded RT primers to viral RNA and pooling samples prior to amplification, but the need for specialized oligo-dT magnetic beads might constitute a separate adoption barrier for this method (*Chappleboim et al., 2021*). Finally, methods have been designed to take advantage of the extreme sensitivity and isothermal conditions of loop-mediated isothermal amplification (LAMP) (*Peto et al., 2021*; *Dao Thi et al., 2020*; *Ludwig et al., 2021*), but they require additional manipulation to introduce barcodes (*Peto et al., 2021*; *Dao Thi et al., 2020*) or do not allow for convenient multiplexing (*Ludwig et al., 2021*).

In this study, we present COV-ID, a method for SARS-CoV-2 identification based on reverse transcription loop-mediated isothermal amplification (RT-LAMP), which enables large-scale diagnostic testing at low cost and with minimal on-site equipment. COV-ID is a robust method that can be used to test tens of thousands of samples for multiple pathogens with modest reagent costs and 2–4 laboratory personnel, generating results within 24 hr. COV-ID uses unpurified saliva or saliva absorbed on filter paper as input material, thus enabling the massively parallel, inexpensive testing required for population-level surveillance of the COVID-19 pandemic (*Figure 1A*).

## Results

### Two-step amplification and indexing of viral and human sequences via RT-LAMP and PCR

The molecular basis for COV-ID is RT-LAMP, an alternative to PCR that has been used extensively for viral DNA or RNA detection in clinical samples (*Li et al., 2011*; *Shirato et al., 2014*; *Calvert et al., 2017*; *Enomoto et al., 2005*), including SARS-CoV-2 (*Augustine et al., 2020*; *United States Food and Drug Administration, 2020*). RT-LAMP requires 4–6 primers that recognize different regions of the target sequence (*Nagamine et al., 2002*; *Notomi et al., 2000*) and proceeds through a set of primed and self-primed steps to yield many inverted copies of the target sequence spanning a range of molecular sizes (*Figure 1—figure supplement 1*). The forward inner primer (FIP) and backward

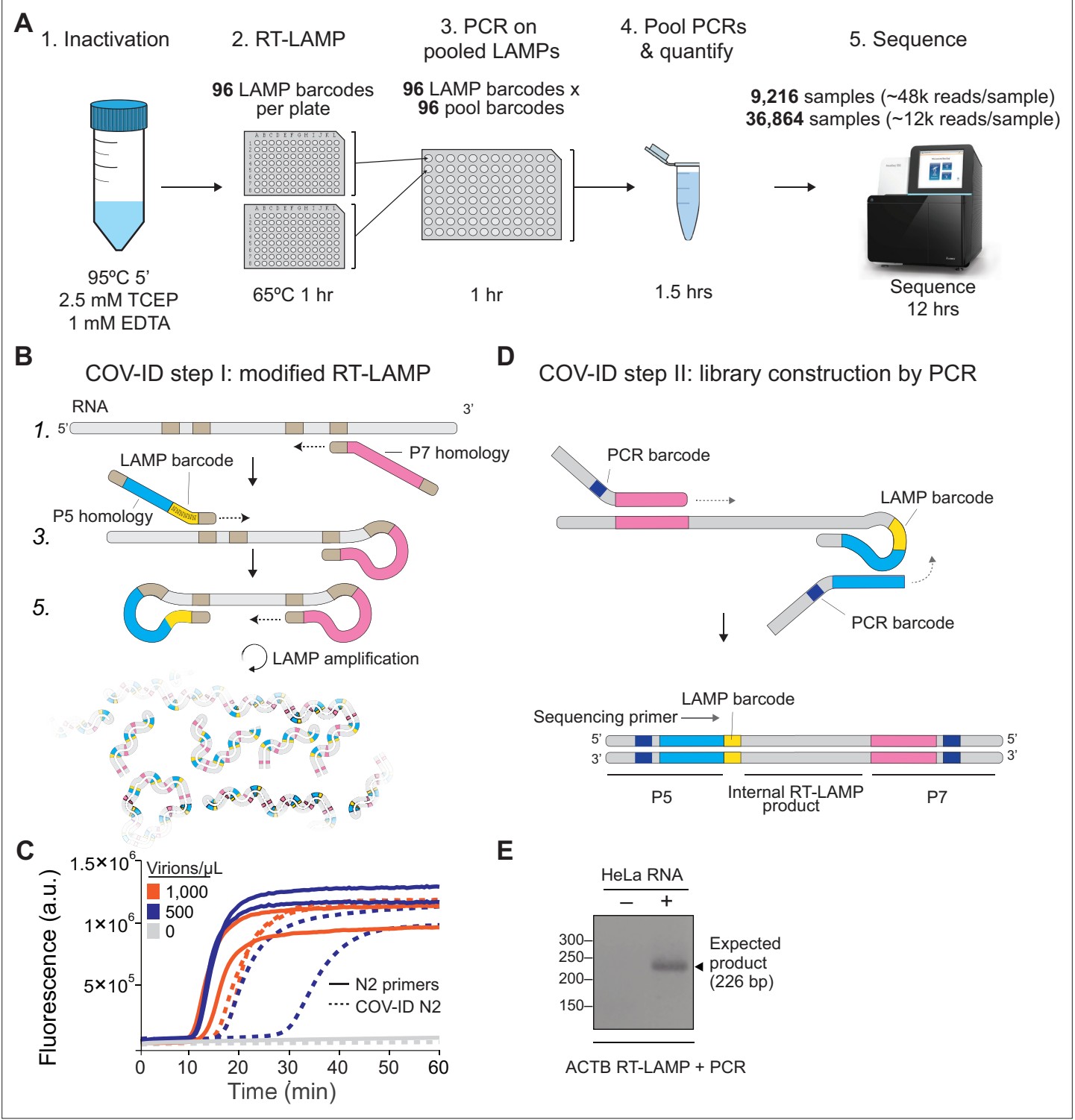

**Figure 1.** Barcoding and PCR amplification of reverse transcription loop-mediated isothermal amplification (RT-LAMP) products. (**A**) Overview of COV-ID. Saliva is collected and inactivated prior to RT-LAMP performed with up to 96 individual sample barcoded primers. LAMP reactions are pooled and further amplified via PCR to introduce Illumina adapter sequences and pool-level dual indexes. A single thermal cycler can amplify 96 or 384 such pools and the resulting 'super-pool' can be sequenced overnight to detect multiple amplicons from 9,216 or 36,864 individual patient samples (number of reads in parenthesis assume an output of ~450 M reads from a NextSeq 500). (**B**) Schematic of the RT-LAMP (step I) of COV-ID. Selected numbered intermediates of RT-LAMP reaction are shown to illustrate how the LAMP barcode, shown in yellow, and the P5 and P7 homology sequences (blue and pink, respectively) are introduced in the final LAMP product. Upon generating the dumbbell intermediate, the reaction proceeds through rapid primed

*Figure 1 continued on next page*

*Figure 1 continued*

and self-primed extensions to form a mixture of various DNA amplicons containing sequences for PCR amplification. A more detailed version of the LAMP phase of COV-ID, including specific sequences, is illustrated in *Figure 1—figure supplement 1*. (**C**) Conventional RT-LAMP primers (solid lines) or primers modified for COV-ID (dotted lines) were used for RT-LAMP of SARS-CoV-2 in saliva. The numbers of inactivated SARS-CoV-2 virions per µL is indicated in the color legend. Each line represents an independent biological replicate. Fluorescence is shown in arbitrary units. (**D**) Schematic of the PCR (step II) of COV-ID. Following RT-LAMP, up to 96 reactions are pooled and purified and Illumina libraries are generated directly by PCR with dual-indexed P5 and P7 adapters in preparation for sequencing. (**E**) COV-ID primers targeting ACTB mRNA were used for RT-LAMP from HeLa total RNA. LAMP was diluted 1:100, amplified via PCR and resolved on 2% agarose gel.

The online version of this article includes the following source data and figure supplement(s) for figure 1:

**Source data 1.** Uncropped blot for *Figure 1E*.

**Figure supplement 1.** Detailed COV-ID mechanism.

inner primer (BIP), which recognize internal sequences, are incorporated in opposite orientation across the target sequence in the final amplified product (*Figure 1—figure supplement 1*).

Previous studies have shown that the FIP and BIP tolerate insertions of exogenous sequences between their different target homology regions (*Yamagishi et al., 2017*). We exploited this flexibility and introduced (1) patient-specific barcodes as shown previously (*Peto et al., 2021*; *Ludwig et al., 2021*; *Yamagishi et al., 2017*) and (2) artificial sequences that allowed for PCR amplification of a small product compatible with Illumina sequencing library construction (*Figure 1*, *Figure 1—figure supplement 1*). This innovation allows us to pool individually barcoded RT-LAMP reactions and amplify them in batch via PCR, while introducing unique P5 and P7 dual indexes in different pools, thus enabling two-dimensional barcoding and dramatically increasing method throughput (see *Supplementary file 1a* for PCR primer sequences). To minimize pool variability, PCR primers can be titrated to 100 nM and pooled PCRs carried out to completion, resulting in each pool being amplified to the same approximate concentration. Uniquely amplified and barcoded pools are mixed into a single 'super-pool' that can be sequenced on an Illumina NextSeq or similar instrument (*Figure 1A*). Combining individual barcodes embedded in the product at the RT-LAMP step with dual indexes introduced at the pool level during the PCR step allows for deconvolution of thousands or tens of thousands of samples in a single sequencing run.

To determine whether introduction of these exogenous sequences into the primers inhibited the isothermal amplification step, we performed RT-LAMP on inactivated SARS-CoV-2 virus using an extensively validated primer set against the N2 region of the nucleocapsid protein (*Butler et al., 2021*) including either the conventional BIP and FIP primers or their modified version re-engineered for the COV-ID workflow (*Figure 1B*). Although the appearance of the amplified viral product was slightly delayed when using COV-ID primers, all reactions reached saturation rapidly and without detectable amplification of negative controls (*Figure 1C*). Next, we tested whether COV-ID was compatible with RT-LAMP using newly designed primers against a host (human) transcript and whether the second step of COV-ID, direct library construction and indexing via PCR amplification (*Figure 1D*), yielded the desired product. For this, we designed RT-LAMP primers against the human beta-actin (*ACTB*) transcript that included sequences necessary for COV-ID. After RT-LAMP, reactions were diluted 100-fold before PCR with barcoded Illumina adapters. A PCR product of the expected size was visible in reactions containing total HeLa RNA, whereas no PCR product was observed in the absence of template (*Figure 1E*). Sanger sequencing of the PCR product confirmed that RT-LAMP followed by PCR generated the product expected by the COV-ID method design, including the sample barcode introduced during the RT-LAMP step.

Thus, our data show that RT-LAMP is tolerant of sequence insertions in the BIP and FIP primers that allow introduction of LAMP-level barcodes as well as sequences homologous to Illumina adapters for direct amplification, indexing, and library construction via PCR.

## Sequencing-based detection of SARS-CoV-2 RNA from saliva using COV-ID

We next evaluated the utility of COV-ID to detect viral RNA in saliva. Saliva is an attractive sample material for COVID-19 diagnostics with potential for early detection (*Savela, 2021*), and has been shown to be a viable template for nucleic acid amplification via RT-PCR (*Ranoa et al., 2020*), recombinase polymerase amplification (RPA) (*Myhrvold et al., 2018*), as well as RT-LAMP (*Lalli et al., 2021*;

*Rabe and Cepko, 2020*). We prepared human saliva for RT-LAMP using a previously described treatment that inactivates SARS-CoV-2 virions, saliva-borne RNases and LAMP inhibitors (*Figure 2A*; *Rabe and Cepko, 2020*). We performed RT-LAMP followed by PCR on inactivated saliva spiked with water or 1000 total copies of inactivated SARS-CoV-2 virus. We observed a single band of the expected size in reactions performed on saliva spiked with virus but not in control reactions (*Figure 2B*). The sequence of the amplified and barcoded viral product was confirmed by Sanger sequencing (*Figure 2—figure supplement 1A*). Next, we subjected the libraries to deep sequencing. Reads aligned uniformly to the *N* gene, the region targeted by the N2 primer set, in COV-ID libraries constructed from viral samples but not in control libraries (*Figure 2C*).

In several SARS-CoV-2 FDA approved tests, parallel amplification of a host (human) amplicon is implemented as a metric for sample integrity and quality. That is, if no human RNA is amplified from a clinical sample, no conclusion can be drawn from a negative test result (*Babiker et al., 2020*). However, in most tests, viral and human amplicons must be detected separately, resulting in a multiplication of the number of reactions to be performed. We reasoned that the deep sequencing nature of COV-ID would allow for simultaneous detection of viral, human, and other control amplicons, without increasing the number of necessary reactions. In fact, given that the PCR handles inserted in the BIP and FIP are the same for all RT-LAMP amplicons (*Figure 1B*), the same P5 and P7 Illumina primers allow the simultaneous amplification of all RT-LAMP products obtained with COV-ID-modified primer sets (*Figure 1D*). To identify a suitable human control, we compared conventional RT-LAMP primers for the mRNA of *ACTB* (*Butler et al., 2021*) or *STATHERIN* (*STATH*), a gene expressed specifically in saliva (*Satoh et al., 2018*). To determine which of the two RT-LAMP primer sets was a better proxy to measure RNA integrity in saliva samples, we assayed for amplification of the respective products in presence or absence of RNase. Whereas addition of RNase A abolished the *STATH* signal, it was ineffectual for *ACTB* (*Figure 2—figure supplement 1B*), suggesting that amplification of genomic DNA made considerable contributions to the RT-LAMP signal observed for the latter. Therefore, we utilized *STATH* mRNA as a human control in subsequent experiments.

We used COV-ID-adapted primer sets for *N2* and *STATH* (*Supplementary file 1a*) in multiplex on inactivated saliva spiked with a range of SARS-CoV-2 from 5 to 10,000 virions/µL. Subsequently, each RT-LAMP reaction was separately amplified via PCR using a unique P5 and P7 index combination, pooled, quantified, and deep-sequenced to an average depth of 6,000 reads per sample. After read trimming, alignment, and filtering (see Materials and Methods), 76% of reads from saliva COV-ID reactions were informative (*Figure 2—figure supplement 1C*). In order to differentiate SARS-CoV-2 positive and negative samples, we calculated the ratio between N2 reads and reads mapping to the human *STATH* control. Using the highest *N2/STATH* read ratio in control (SARS-CoV-2 negative saliva) as a threshold, 95% (19/20) of samples with spiked-in virus were correctly classified as positives (*Figure 2D*). Using COV-ID, we consistently detected SARS-CoV-2 in saliva samples containing as low as 5 virions per µL, a limit of detection comparable and in some cases superior to those of established testing protocols (*MacKay et al., 2020*).

Scaling COV-ID to handle higher sample numbers requires pooling samples immediately following RT-LAMP, prior to the PCR step (*Figure 1A*). We designed 32 unique 5-nucleotide barcodes for several target LAMP amplicons (*Figure 2—figure supplement 1D* and *Supplementary file 1b*). We first individually validated each barcode and primer combination by real-time fluorescence and PCR efficiency. Certain barcodes inhibited the RT-LAMP reaction, possibly due to internal micro-homology and primer self-hybridization (*Torres et al., 2011*). Nonetheless, out of 32 barcodes tested in 3 separate RT-LAMP reactions (*N2*, *ACTB*, and *STATH*), 25 successfully amplified all three target RNAs (*Figure 2—figure supplement 1D*). Saliva samples spiked with various concentrations of inactivated SARS-CoV-2 were amplified via barcoded RT-LAMP, then optionally pooled prior to PCR and sequencing (*Figure 2—figure supplement 1E*). SARS-CoV-2/STATH ratios demonstrated no loss of sensitivity or specificity in the pooled samples compared to the individual PCRs.

To test the potential of COV-ID on patient samples, we tested saliva specimens collected and previously analyzed at the Hospital of the University of Pennsylvania (see Materials and Methods). We carried out multiplex barcoded RT-LAMPs on each sample (COV-ID step I, *Figure 1B*), pooled the reactions and then constructed libraries via PCR (COV-ID step II, *Figure 1D*). After deep sequencing, analysis of *N2/STATH* ratios showed 100% (8/8) concordance with viral copy numbers generated by

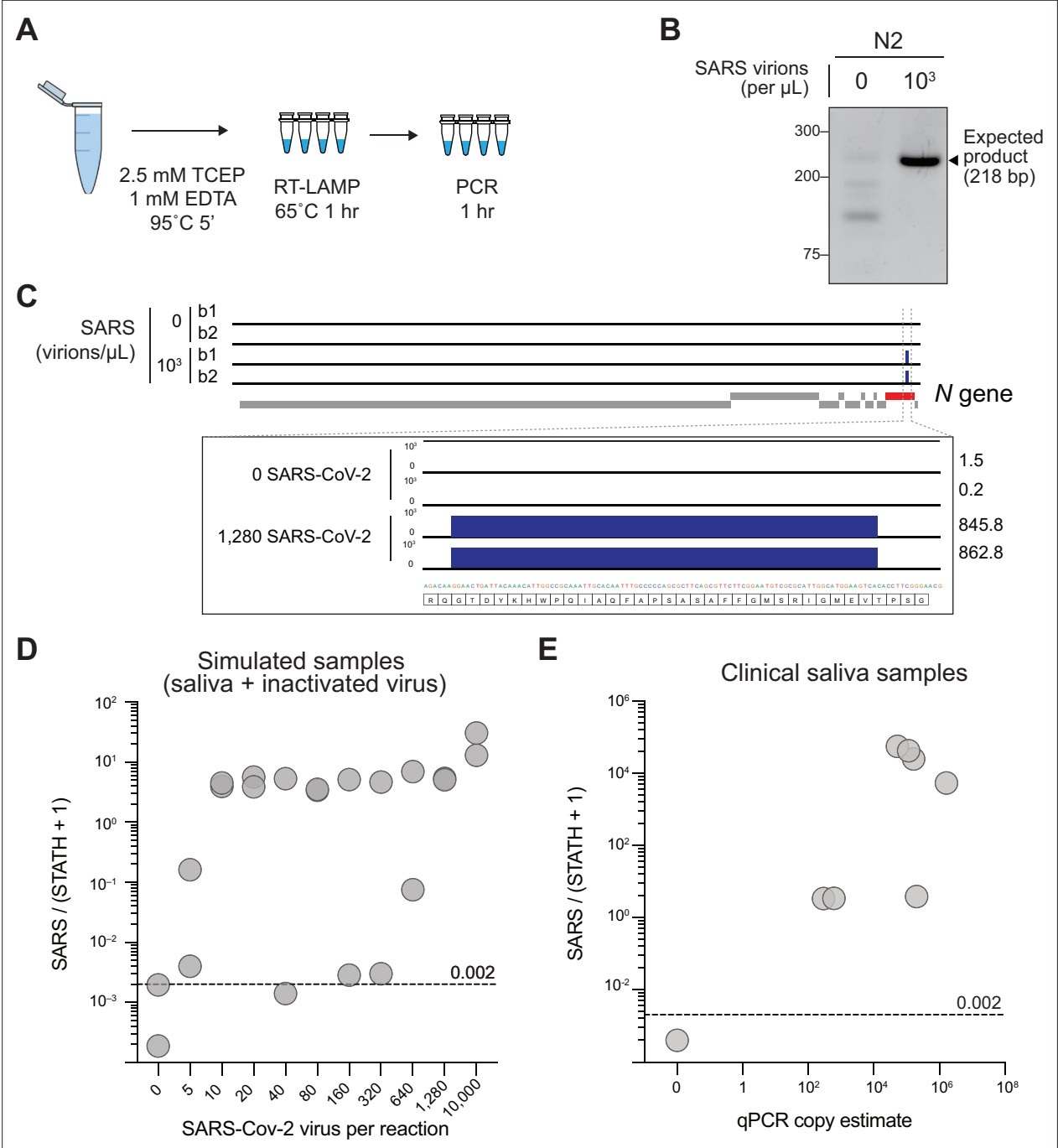

**Figure 2.** Sequencing-based detection of SARS-CoV-2 in saliva samples. (**A**) Saliva preparation. Crude saliva was inactivated via TCEP/EDTA addition and 95 °C incubation prior to RT-LAMP. (**B**) RT-LAMP followed by COV-ID PCR performed directly on saliva. Saliva with and without addition of 1,000 copies of inactivated SARS-CoV-2 templates was inactivated as described in (**A**), then used as template. (**C**) Alignment of sequenced reads against SARS-CoV-2 genome from COV-ID of inactivated saliva spiked with or without 1,280 virions SARS-CoV-2 per μL. All SARS-CoV-2 reads align exclusively to expected region of the N gene. Open reading frames of viral genome are depicted via gray boxes below alignment. Inset: scale shows reads per 1,000. Height of peak is provided on the right. (**D**) Scatter plot for the ratio of SARS-CoV-2 / (*STATH* +1) reads obtained by COV-ID (y axis) versus the number of virions per μL spiked in human saliva (x axis). The threshold was set above the highest values scored in a negative control (dashed line). Each circle represents an independent biological replicate. (**E**) COV-ID performed on clinical saliva samples. The scatter plot shows the SARS-CoV-2 / (*STATH* +1) read ratio (y axis) versus the viral load in the sample estimated by a clinically approved, qPCR-based diagnostic test. The threshold was set based on the negative controls shown in (**D**). Each circle represents an independent biological replicate.

The online version of this article includes the following source data and figure supplement(s) for figure 2:

*Figure 2 continued on next page*

*Figure 2 continued*

**Source data 1.** Uncropped blot for *Figure 2B*.

**Figure supplement 1.** Optimization of COV-ID in human saliva.

**Figure supplement 2.** Reverse transcription loop-mediated isothermal amplification (RT-LAMP) on a SARS-CoV-2 synthetic calibration standard.

**Figure supplement 3.** Clinical validation of COV-ID on RNA from nasopharyngeal (NP) swabs.

a standard clinical test (RNA purification followed by RT-qPCR) (*Figure 2E*), demonstrating the effectiveness of the COV-ID approach.

## Calibration of COV-ID using an internal standard

Existing deep sequencing approaches for massively parallel COVID-19 testing based on RT-PCR incorporate artificial spike-ins, which serve as an internal calibration controls and allow for better estimates of viral loads by end-point PCR (*Bloom et al., 2021*; *Yelagandula et al., 2021*). At the same time, adding to the reactions an artificial substrate for amplification helps minimizing spurious signals as it can 'scavenge' viral amplification primers in negative samples. Finally, by providing a baseline amplification even in empty samples, a strategy designed to use an internal calibration control can reduce variance in total amounts of final amplified products across samples, which compresses the dynamic-range of sequence coverage for each patient in a complex pool and, therefore, reduces the risk of inconclusive samples due to low sequencing coverage (*Yelagandula et al., 2021*).

We reasoned that an internal calibration control approach for LAMP-based quantification would provide similar benefits in the context of COV-ID. To generate a SARS-CoV-2 synthetic calibration standard, we synthesized a fragment of the *N2* RNA that retained all primer-binding regions for RT-LAMP and contained a divergent 7-nt stretch of sequence to distinguish reads originating from the calibration standard from those originating from the natural virus (*Figure 2—figure supplement 2A*). After confirming that the calibration control template was efficiently amplified via RT-LAMP with the *N2* primer set (*Figure 2—figure supplement 2B*), we performed pooled COV-ID on virus-containing saliva in the presence of 20 fg of *N2* synthetic calibration control RNA. As expected (*Yelagandula et al., 2021*), addition of a constant amount of an internal calibration control across reactions reduced the variability in total read numbers for individual samples in the final pool (*Figure 2—figure supplement 2C*). As discussed above, a narrower range in sequencing output across samples in a pool optimizes the utilization of sequencing reads, and ultimately lowers the cost per sample. Including the internal calibration control in the normalization strategy resulted in lower levels of false positive signal from negative samples (*Figure 2—figure supplement 2D*). This is likely because in cases where very few *STATH* reads were obtained, possibly due to degradation of host RNA in the saliva sample, the resulting small denominator inflated the *N2*/*STATH* ratio even for SARS-CoV-2 signal that was low in absolute terms and likely spurious.

Thus, these data show that synthetic standards are compatible with the COV-ID workflow and provide a means to stabilize total amplification and read allocation per sample, while also offering an additional calibration control to better estimate the viral load in samples where the endogenous *STATH* mRNA might be below detection due to improper collection or handling.

## Validation of COV-ID on clinical nasopharyngeal swab samples

To evaluate the scalability of COV-ID, we analyzed 120 clinical RNA specimens purified from NP swab samples collected from patients at the Hospital of the University of Pennsylvania. Each barcoded RT-LAMP reaction (COV-ID step I) was grouped in pools of 10 samples per PCR amplification (COV-ID step II), allowing us to estimate the feasibility of the two-dimensional barcoding strategy. Again, COV-ID was in good agreement with quantification of viral RNA via a clinical RT-qPCR assay (*Figure 2—figure supplement 3A*) and was able to distinguish positive from negative patient samples in the same RT-LAMP pool (*Figure 2—figure supplement 3B*). COV-ID showed inferior sensitivity compared to individual RT-qPCRs, failing to detect 10 samples that displayed signal by qPCR. However, 7 of these 10 'false negatives' had very high Ct values (higher than 36 cycles), which are unreliable and poorly reproducible, even when using state-of-the-art TaqMan qPCR (*Yelagandula et al., 2021*), and could not be classified properly by another sequencing-based high-throughput method (*Yelagandula et al., 2021*).

Overall, on these 120 clinical samples, COV-ID demonstrated good specificity and sensitivity, as shown by receiver operator characteristic and precision-recall curves, especially when only samples with Ct values < 36 or < 31 were considered 'true positives' (*Figure 2—figure supplement 3C, D*; middle and right panels, respectively).

Taken together, our data show that COV-ID can be utilized to detect viral and human amplicons in multiplex from contrived and clinical samples.

## Simultaneous detection of SARS-CoV-2 and influenza A by COV-ID

Given the challenge of distinguishing early symptoms of COVID-19 from other respiratory infections, we evaluated COV-ID for the simultaneous detection of more than one viral pathogen. Multiple distinct products can be simultaneously amplified by RT-LAMP in the same tube by providing the appropriate primer sets in multiplex, as we demonstrated above by co-amplifying *N2* and *STATH* in the same COV-ID reaction (see *Figure 2*). In fact, simultaneous detection of SARS-CoV-2 and influenza virus by RT-LAMP was previously achieved, albeit in a fluorescent-based, low-throughput assay (*Zhang and Tanner, 2021*). We reasoned that the sequencing-based readout of COV-ID would allow extending this approach to the simultaneous detection of multiple pathogens as well as endogenous (host mRNA) and synthetic calibration standards, all in a single reaction.

To test the ability of COV-ID to simultaneously detect multiple viral templates, we selected and validated a generic 'flu' RT-LAMP primer set that recognizes several strains, including influenza A virus (IAV) and influenza B (*MacKay et al., 2020*; *Takayama et al., 2019*), and modified the BIP and FIP sequence to introduce the COV-ID barcodes and handles for PCR (*Figure 2—figure supplement 1D* and *Supplementary file 1a*). We added inactivated SARS-CoV-2 virus (BEI resources) and IAV strain H1N1 RNA (Twist Biosciences) to saliva according to a $3 \times 4$ matrix of ($10^4$, $10^3$, or 0 copies per µL) SARS-CoV-2 RNA against H1N1 RNA ($10^5$, $10^4$, $10^3$, or 0 copies per µL) (*Figure 3A*), as well as the *N2* synthetic calibration standard. We performed multiplex COV-ID on these samples using primers sets for *STATH*, *N2* (to detect SARS-CoV-2), and IAV (to detect H1N1) and sequenced to an average depth of 21,000 reads per sample. Both H1N1 and SARS-CoV-2 were detected above background and the signal correlated with the amount of the respective template added to saliva (*Figure 3B, C*). Overall, multiplex COV-ID correctly identified samples that contained only SARS-CoV-2 (7/8) or H1N1 (6/8). For samples that contained both pathogens we observed reduced sensitivity (11/16 identification of both pathogens), which was also observed in a previous multiplexing attempt (*MacKay et al., 2020*). However, in practice individuals who are simultaneously infected with both viruses presumably would be rare, and for these cases the ability to detect at least one virus successfully would allow to follow up with further diagnostic testing. We found that of the samples containing both viruses, 16/16 showed positive detection of at least one pathogen (SARS-CoV-2 or H1N1), suggesting the reduced sensitivity of the multiplex assay is due to interference between amplification of the two viral templates. This also demonstrates that COV-ID can be used as an effective screening approach for multiple viral templates.

## Paper-based saliva sampling for COV-ID

As an additional step toward increasing the throughput of the COV-ID approach, we explored avenues to simplify collection, lower costs, and expedite processing time. Absorbent paper is an attractive alternative to sample vials for collection, given its low cost, wide availability, and smaller environmental footprint. In fact, paper has been used as a means to isolate nucleic acid from biological samples for direct RT-PCR testing (*Glushakova et al., 2017*) as well as RT-LAMP (*Kellner et al., 2020*; *Yaren et al., 2021*).

We sought to determine whether the COV-ID workflow would be compatible with saliva collection on absorbent paper. First, we immersed a small square of Whatman filter paper into water containing various dilutions of inactivated SARS-CoV-2. After 2 min, the paper was removed and transferred to PCR strip tubes followed by heating at 95°C for 5 min to air-dry the sample (*Figure 4A*). Next, we added the RT-LAMP mix containing the *N2* COV-ID primer set directly to the tubes containing the paper squares and let the reaction proceed in the usual conditions. COV-ID PCR products of the correct size were evident in all samples containing viral RNA, with sensitivity of at least 100 virions/µL (*Figure 4B*) and in none of the controls, demonstrating that the presence of paper does not interfere with the RT-LAMP reaction and subsequent PCR amplification with Illumina adapters.

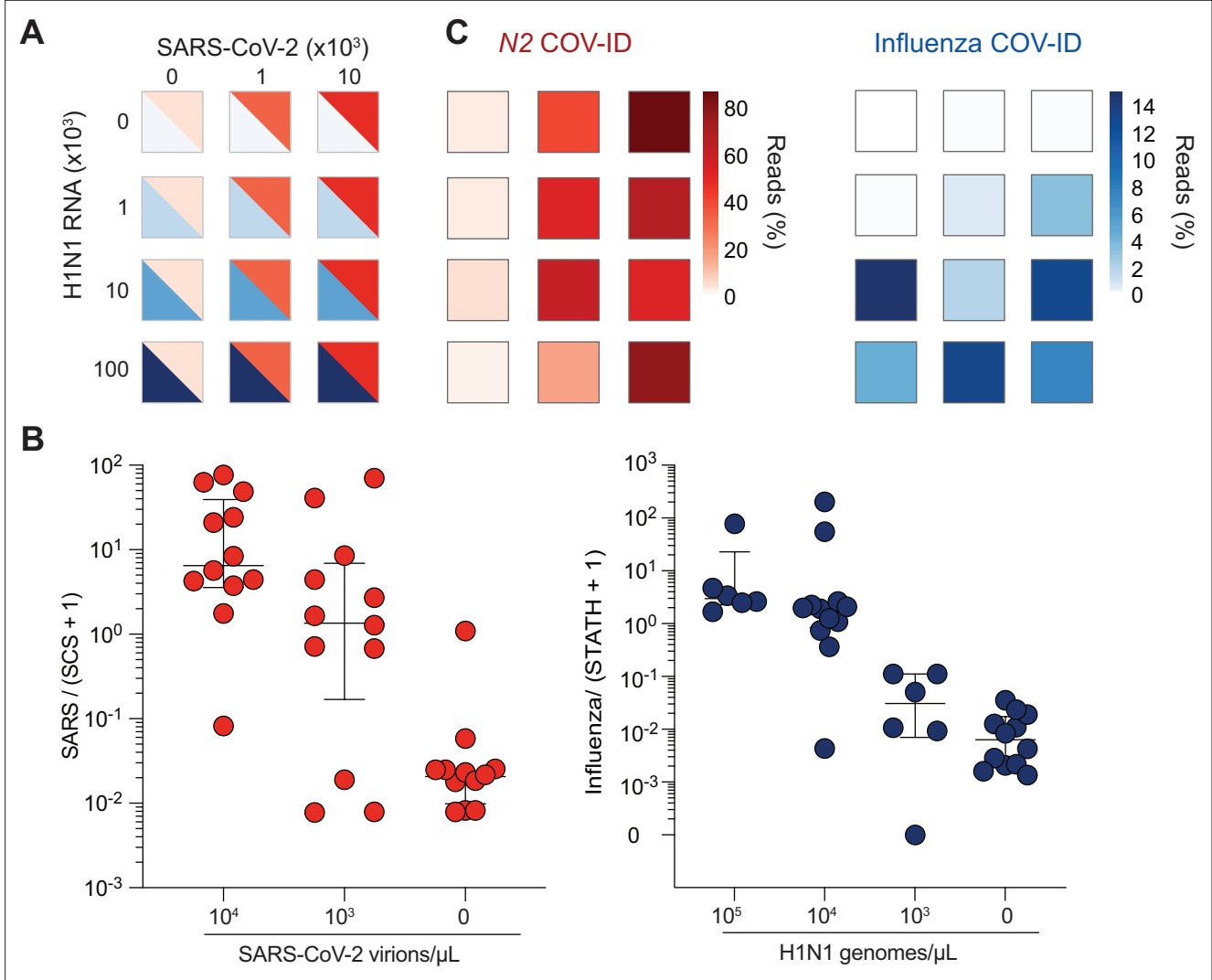

**Figure 3.** COV-ID multiplex detection of SARS-COV-2 and Influenza A. (**A**) TCEP/EDTA treated saliva was spiked with indicated amounts of BEI heat-inactivated SARS-CoV-2 or H1N1 influenza A RNA to the indicated concentration of virions/genomes per μL. One microliter of saliva was used for COV-ID reactions. (**B**) COV-ID was performed in two independent experiments on saliva samples from the matrix shown in (**A**) in the presence of 20 femtograms N2 synthetic calibration standard (SCS) using N2, influenza (**MacKay et al., 2020**) and STATH COV-ID primers. N2/(SCS +1) and influenza/(STATH +1) read ratios are displayed with bars showing median ± interquartile range. Circles represent individual biological replicates. Samples were considered positive for a given sequence if the associated read ratio was greater than 2x the maximum value in the control saliva samples. (**C**) Heatmaps of SARS-CoV-2 (left) or H1N1 (right) COV-ID signal in multiplex reaction. Individual data points are from (**B**). The heatmap color represents the mean of the percentage of viral reads in each sample.

To assay direct COV-ID detection from saliva on paper, we saturated Whatman filter paper squares with saliva containing different amounts of inactivated SARS-CoV-2 virus, which, we reasoned, would be equivalent to a patient collecting their own saliva by chewing on a small piece of absorbent paper. Next, we placed the paper squares into reaction tubes containing TCEP/EDTA inactivation buffer (see Materials and Methods) similar to that used for the in-solution samples used in our previous experiments (see *Figure 1A*). We dried the paper at 95°C and performed RT-LAMP followed by PCR (*Figure 4C*), which resulted in the appearance of COV-ID products of the correct size starting from saliva spiked with as few as 50 virions/μL (*Figure 4D*). We then performed COV-ID sequencing on saliva collected on paper using primers N2 and *STATH* in the presence of the *N2* synthetic calibration standard RNA. The sequence data showed more variability and limited coverage of the control amplicons compared to in-solution COV-ID, likely due to the more challenging reaction conditions; therefore, we normalized viral reads using both *STATH* reads and synthetic calibration standard reads. This

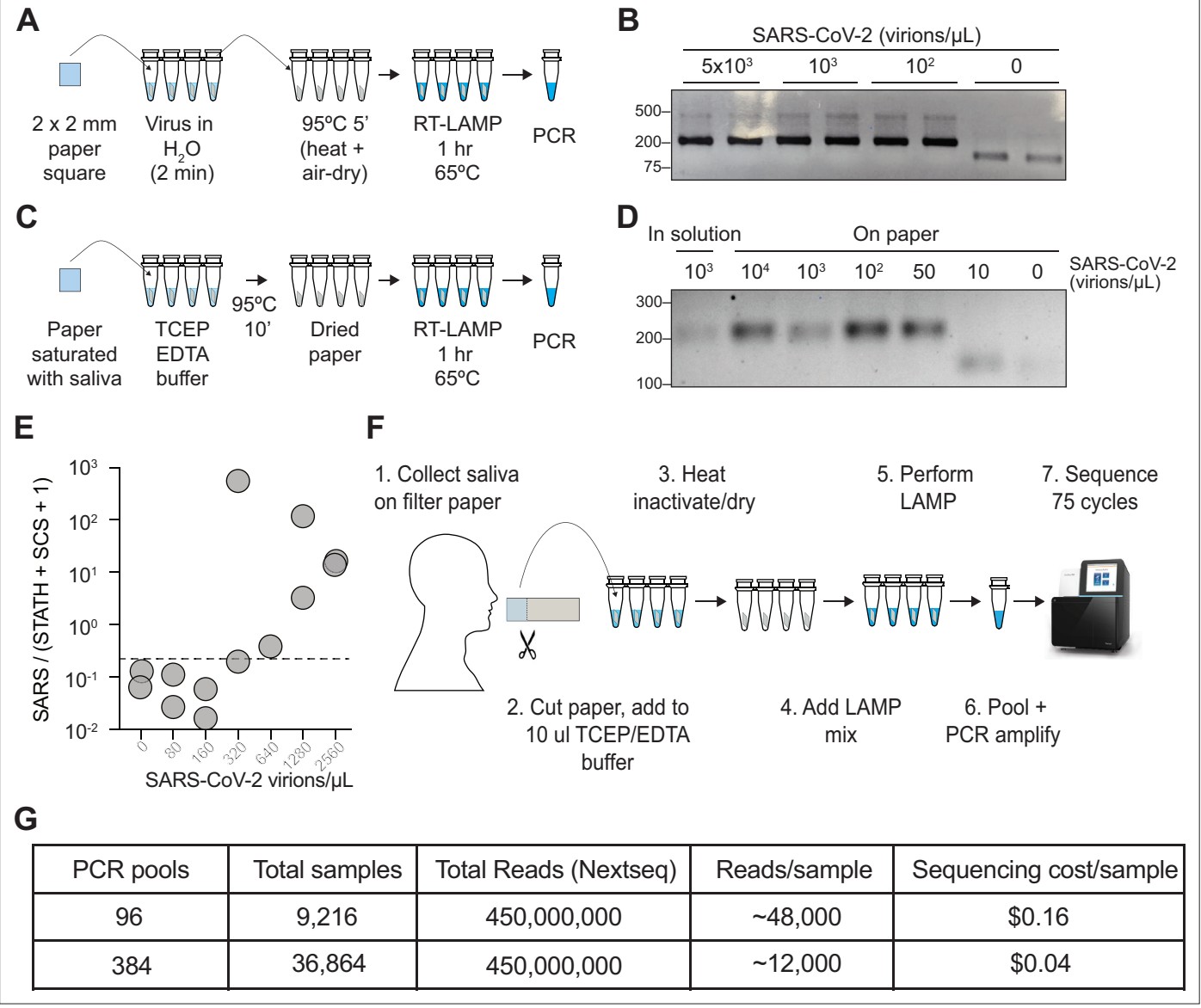

**Figure 4.** COV-ID on saliva collected on paper. (**A**) Scheme for COV-ID on viral RNA absorbed on paper. (**B**) PCR reactions from paper samples immersed in water with indicated viral concentrations then amplified with N2 COV-ID primers. (**C**) Scheme for COV-ID on saliva spiked with viral and RNA and absorbed on paper. (**D**) Same as (**B**) but on saliva absorbed on paper. (**E**) SARS-CoV-2 virus was added to saliva and prepared as in (**C**).Reverse transcription loop-mediated isothermal amplification (RT-LAMP) and sequencing was carried out in presence of calibration standard RNA. Viral reads are presented as ratio against the sum of STATH and N2 synthetic calibration standard (SCS) reads. Positive threshold was set as 2x maximum value in negative saliva and indicated by dashed horizontal line. (**F–G**) Paper-based COV-ID workflow (**F**) and cost calculations (**G**). Saliva is collected orally on a precut strip of paper, from which a 2 mm square would be cut out and added to a reaction vessel containing TCEP/EDTA inactivation buffer and processed as shown in (**C**).

The online version of this article includes the following source data for figure 4:

**Source data 1.** Uncropped blot for *Figure 4B*.

**Source data 2.** Uncropped blot for *Figure 4D*.

paper-based COV-ID proof-of-principle experiment detected the presence of viral RNA in samples with as little as 320 copies/µL (*Figure 4E*), a higher limit of detection compared to that of in-solution COV-ID, but still well within the useful range (*Winnett et al., 2020*) to detect infections.

Taken together, these data show that the RT-LAMP step of COV-ID is compatible with the presence of paper in the reaction tube and suggest that self-collection of saliva by patients directly on

absorbent paper could provide a simple and cost-effective strategy to test thousands of saliva samples for multiple pathogens (*Figure 4F*).

## Discussion

Testing strategies are vital to an effective public health response to the COVID-19 pandemic, particularly with the spread of the disease by asymptomatic individuals. An ongoing challenge to COVID-19 testing is the need for massive testing strategies for population-level surveillance that are needed for efficient contact tracing and isolation. Many FDA-approved clinical SARS-CoV-2 diagnostic tests are based on time-consuming and expensive protocols that include RNA purifications and RT-PCR and must be performed by trained personnel in well-equipped laboratories (*MacKay et al., 2020*; *Woronik et al., 2021*). Point-of-care antigen tests provide a much faster turnaround time and require little manipulation, but they lack in sensitivity compared to tests that detect viral RNA (*Brümmer et al., 2021*). Because of reagent limitations and diagnostic testing bottlenecks, at the beginning of the pandemic symptomatic individuals and individuals particularly vulnerable for infection after exposure were prioritized for diagnostic COVID testing (*Schuetz et al., 2020*). Private organizations, including colleges and universities, circumvented some of these challenges by contracting with private laboratories to establish asymptomatic surveillance testing protocols; however, this is a costly option for population-level surveilling of asymptomatic SARS-CoV-2 infections.

Several effective COVID-19 vaccines have been developed and there is a concerted ongoing global vaccination effort, providing a concrete means to end the pandemic. Despite this progress, there are several potential risks that require vigilance: possible COVID-19 transmission in vaccinated individuals, emergence of vaccine-resistant viral variants, and public skepticism of vaccines or faltering compliance with social distancing guidelines (*Aschwanden, 2021*). For these reasons ongoing testing and surveillance efforts will remain important for the foreseeable future, both to monitor the progress of vaccination in reducing symptomatic cases and to detect emerging variants.

In order to scale testing to an effective volume and frequency, surveillance tests must demonstrate the following qualities: (1) <u>sensitivity</u>, to identify both asymptomatic and symptomatic carriers; (2) <u>simplicity</u> in methodology, to be performed in a number of traditional diagnostic laboratories, without specialized equipment; (3) <u>low cost</u> and easily accessible reagents; (4) <u>ease of collection</u> method; (5) <u>rapid</u> turnaround time to allow for isolation and contract tracing; and (6) ability to <u>co-detect</u> multiple respiratory viruses, given the overlap in patient symptoms. To this end, we have developed COV-ID, an RT-LAMP-based parallel sequencing SARS-CoV-2 detection method that can provide results from tens of thousands of samples per day at relatively low cost to simultaneously detect multiple respiratory viruses.

COV-ID features key innovations that make it well-suited to high-throughput testing. First, COV-ID uses a two-dimensional barcoding strategy (*Yelagandula et al., 2021*), where the same 96 barcodes are used in each RT-LAMP plate, making it possible to pre-aliquot barcodes in 96-well plates ahead of time and store them at –20°C, simplifying execution of the assay and shortening turnaround times. Second, since RT-LAMP does not require thermal cycling, tens of thousands of samples can be run simultaneously in a standard benchtop-sized incubator or hybridization oven held at 65°C. Third, individual samples are pooled immediately following RT-LAMP; therefore, a single thermocycler has the potential to process up to 96 or 384 RT-LAMP plates, generating 9,216 or 36,864 individually barcoded samples, respectively (*Figures 1A and 4F, G*). Only 96 unique FIP barcodes are required for this scaling; here, we show that 28 out of 32 LAMP barcodes tested were functional for both *N2* and *STATH*. This proof-of-principle experiment demonstrates the feasibility of generating the library of barcodes required to apply COV-ID to a large population. An additional advantage of sequencing-based approaches, such as COV-ID is that with carefully designed primers it would be possible to recover information about viral variants directly from the sequencing reads (*Everett et al., 2021*). Recently, another group has used RT-LAMP coupled with molecular beacons to amplify the spike sequences of SARS-CoV-2 and identify the emerging B.1.1.7 variant (*Sherrill-Mix et al., 2021b*), demonstrating the utility of RT-LAMP to capture variant-specific sequence information. Finally, COV-ID can generate ready-to-sequence libraries directly from saliva absorbed onto filter paper, which would allow for major streamlining of the often-challenging logistical process of sample collection (*Figure 4*). Thus, COV-ID libraries for thousands and tens of thousands of samples can be generated with relatively minimum effort in biological laboratories with basic equipment and easily accessible reagents.

With the average throughput of an Illumina NextSeq 500/550, a relatively affordable next-generation sequencer up to 9,216 (96 RT-LAMPs x 96 pools) can be sequenced at a depth of ~48,000 reads per sample, and up to 36,864 (96 RT-LAMPs x 384 pools) can be sequenced at a depth of ~12,000 reads, which, we showed, is more than sufficient to obtain information about multiple viral and control amplicons. Considering that reagents for one NextSeq run cost ~1,500 U.S. dollars, the theoretical sequencing cost per sample could be as low as $0.04 (*Figure 4G*). While sequencing instruments are relatively specialized and not ubiquitous, amplified COV-ID DNA libraries could be shipped to remote facilities for sequencing in a cost-effective manner as previously proposed by the inventors of LAMP-seq (*Ludwig et al., 2021*). In a context where a sequencing instrument is available locally, with optimized sample collection and streamlined logistics results could be turned around within 16 hr. Finally, because of the limited sequence space against which reads must be aligned, computational analysis of the resulting data can be performed in a matter of minutes with optimized pipelines, providing results shortly after the sequencing run has completed.

Several methods have emerged that harness massively parallel next generation sequencing for diagnostics of SARS-CoV-2 (*Simonetti et al., 2021*; *Bloom et al., 2021*; *Yelagandula et al., 2021*; *Aynaud et al., 2021*; *Wu et al., 2021*; *de Mello Malta et al., 2021*; *Chappleboim et al., 2021*; *Peto et al., 2021*; *Dao Thi et al., 2020*; *Ludwig et al., 2021*, *Credle et al., 2021*), reflecting the desire for novel approaches to address the shortcomings of labor-intensive individual clinical diagnostic testing. COV-ID complements these approaches by providing a method that can screen thousands of individuals with a heated incubator, a single PCR thermocycler and access to a sequencer. COV-ID and existing methods for sequence-based diagnostics each have their strengths and weaknesses and should be carefully evaluated for their suitability in specific cases. While our method demonstrates promise, we note that there remain some points that require optimization prior to successful large-scale application of COV-ID in a population setting, including developing primer sets to sequence the spike of SARS-CoV-2 for use in genomic surveillance of emerging variants, or validating individual RT-LAMP barcodes to ensure optimal throughput.

COV-ID has a limit of detection of 5–10 virions of SARS-CoV-2 per µL in contrived saliva samples (*Figure 2D*) and at least 300 virions/µL in saliva collected from patients in a clinical setting (*Figure 2E*). Of note, the limit of detection in clinical saliva specimens is likely lower, but it could not be determined because the lowest viral load of all our positive samples was 300 virions/µL. Importantly, this was also the apparent limit of detection of paper-based COV-ID (*Figure 4E*). The average viral load in saliva for a contagious individual (the target of population-scale surveillance) is still a subject of debate and likely depends on several factors including virus variant. In one study, RT-qPCR-based estimates of SARS-CoV-2 viral loads averaged 380 copies/µL in saliva samples of infected individuals (*Wyllie et al., 2020*). Some meta analyses indicated no significant difference in viral load at the upper respiratory tract for symptomatic *vs.* asymptomatic individuals (*Walsh et al., 2020*; *Zuin et al., 2021*), and levels of viral RNA in saliva are in general comparable to those found in NP swabs (*Butler-Laporte et al., 2021*). This would suggest that the limit of detection of COV-ID matches and exceeds what is needed to detect viral RNA in saliva of infectious and potentially contagious individuals.

In conclusion, COV-ID is a flexible platform that can be executed at varying levels of scale with additional flexibility in sample input, making it an attractive platform for surveillance testing. Population-level monitoring of SARS-CoV-2 infections will be critical while vaccines are being distributed to the global population, and continued surveillance will likely remain an effective strategy to protect immunocompromised and unvaccinated members in society and within entities and organizations where regular monitoring is critical to social isolation strategies. To that end, effective, low-cost, multiplexed, and readily implementable strategies for surveillance testing, such as COV-ID, are important to mitigate the effects of the current and future pandemics.

## Materials and methods
### RT-LAMP primer design
Primers against *ACTB* were designed using PrimerExplorerV5 (https://primerexplorer.jp/e/) using default parameters and including loop primers (*Supplementary file 1a*).

For COV-ID, priming sequences for PCR were inserted in FIP and BIP primers between the target homology regions (F1c and F2, and B1c and B2, respectively, see *Figure 1—figure supplement 1*).

After testing, we determined that 12 nts and 11 nts were most effective for the P5 and P7 binding regions, respectively, being the shortest insertion that allowed reliable PCR amplification from LAMP products without impacting LAMP efficiency. In addition, a 5 nt barcode sequence was inserted at the immediate 3′ end of the P5-binding region of the FIP primer (see below).

## LAMP barcode design

Starting from the total possible 1,024 unique 5-nt barcodes, we filtered 404 with exact homology to the reverse complement of any RT-LAMP primer used in this study (*N2, STATHERIN, ACTIN,* and influenza). Out of the 620 remaining barcodes, we selected a set of 32 with Hamming distance ≥ 2 between each barcode and all other barcodes of the set. These barcodes were tested in each RT-LAMP primer set using 1,000 copies of the target amplicon set, to determine whether they interfered with the reaction. Primers that failed to show LAMP signal by real time fluorescence monitoring or generate expected PCR product were discarded (*Figure 2—figure supplement 1D*). Final usable barcodes are provided in *Supplementary file 1b*.

## Saliva preparation

We prepared 100× TCEP/EDTA buffer (250 mM TCEP, 100 mM EDTA, and 1.15 N NaOH) (*Rabe and Cepko, 2020*). TCEP/EDTA buffer was added to human saliva at 1:100 volume, then samples were capped, vortexed to mix and heated in a thermocycler (95°C 5 min, 4°C hold) until ready to use for RT-LAMP. When indicated, heat-inactivated SARS-CoV-2 (BEI Resources Cat. NR-52286) or H1N1 genomic RNA (Twist Biosciences Cat. 103001) was added to inactivated saliva prior to RT-LAMP.

## *N2* synthetic calibration standard

To prepare the in vitro transcription template for SARS-CoV-2 *N2* synthetic calibration standard RNA, we performed RT-PCR using Power SYBR RNA-to-Ct kit (Thermo Cat. 4389986) of heat inactivated SARS-CoV-2 (BEI Resources Cat. NR-52286) using the following primers: N2-B3 and N2-spike-T7 S. PCR product was purified and used as a template for in vitro transcription using HiScribe T7 transcription kit (NEB Cat. E2040S). RNA was purified with Trizol (Thermo Cat. 15596026), quantified via $A_{260}$, then aliquoted in BTE buffer (10 mM bis-tris pH 6.7, 1 mM EDTA) and stored at –80°C. The primers used and the sequence of the synthetic calibration standard are provided in *Supplementary file 1a*.

## RT-LAMP

All RT-LAMP reactions were set up in clean laminar flow hoods and all steps before and after LAMP were carried out in separate lab spaces to avoid contamination. RT-LAMP reactions were set up on ice as follow: for each amplicon 5 or 6 LAMP primers were combined into 10× working stock at established concentrations: 16 µM FIP, 16 µM BIP, 4 µM LF, 4 µM LB, 2 µM F3, and 2 µM B3. For multiplexed COV-ID reactions 10× working primer mixes for each amplicon were either added proportionally so that the total primer content remained constant, or mixed so that BIP and FIP primers were scaled down depending on amplicon number while remaining primers (LF and/or LB, F3, B3) were kept at same concentration as in single reactions.

Each 10 µL RT-LAMP reaction mix consisted of 1× Warmstart LAMP 2× Master Mix (NEB Cat. E1700S), 0.7 µM dUTP (Promega Cat. U1191), 1 µM SYTO-9 (Thermo Cat. S34854), 0.1 µL Thermolabile UDG (Enzymatics Cat. G5020L), 1 µL of saliva template, and optionally 20 fg of N2 Spike RNA. For RT-LAMP of purified RNA from NP swabs, 2 µL template in 10 µL reaction volume was used. Reactions were prepared in qPCR plates or eight-well strip tubes, sealed, vortexed and centrifuged briefly, then incubated in either a QuantStudio Flex 7 or StepOnePlus instrument (Thermo) for 65°C 1 hr. Real-time fluorescence measurements were recorded every 30 s to monitor reaction progress but were not used for data analysis. Following LAMP the reactions were heated at 95°C 5 min to inactivate LAMP enzymes.

## Library construction by PCR amplification

All post-LAMP steps were carried out on a clean bench separate from LAMP reagents and workspace. For individual LAMP samples, LAMP amplicons were diluted either 1:100 or 1:1,000 in water. For pooling of individually barcoded LAMP reactions, equal amounts of all LAMP reactions were combined and then either diluted 1:1000 or purified via SPRIselect beads (Beckman Coulter Cat.

B23317) using a bead-to-reaction ratio of 0.1×. Purified material was diluted to final 100-fold dilution relative to LAMP.

One microliter of diluted LAMP material was used as a template for PCR using OneTaq DNA polymerase (NEB Cat. M0480L) with 100 nM each of custom dual-indexed Illumina P5 and P7 primers in either 10 or 25 μL reaction (*Supplementary file 1a*). PCR reactions were incubated as follows: (25 cycles of stage 1 [94°C × 15 s, 45°C × 15 s, 68°C × 10 s], 10 cycles of Stage 2 [ 94°C × 15 s, 68°C × 10 s], 68°C × 1 min, 4°C × ∞). Note, for initial pilot COV-ID and clinical sample experiments (*Figure 2D–E*, *Figure 2—figure supplement 1C*) PCR incubation was performed as above with modification: [Stage 1 × 10 cycles, Stage 2 × 25 cycles].

PCR products were resolved on 2% agarose gel to confirm library size, then all were pooled and purified via MinElute PCR purification kit (Qiagen Cat. 28004) and quantified using either Qubit dsDNA High Sensitivity kit (Thermo Cat. Q32851) or Kapa Library Quantification Kit for Illumina (Kapa Cat. 07960140001).

## Patient samples

Clinical saliva samples were obtained and characterized as part of a separate study at the University of Pennsylvania (*Sherrill-Mix et al., 2021a*) and collected under Institutional Review Board (IRB)-approved protocols (IRB protocol #842,613 and #813913). Briefly, salivary samples were collected from possible SARS-CoV-2 positive patients at one of three locations: (1) Penn Presbyterian Medical Center Emergency Department, (2) Hospital of the University of Pennsylvania Emergency Department, and (3) Penn Medicine COVID-19 ambulatory testing center. Inclusion criteria including any adult (age >17 years) who underwent SARS-CoV-2 testing via standard nasopharyngeal swab at the same visit. Patients with known COVID-19 disease who previously tested positive previously were excluded. After verbal consent was obtained by a trained research coordinator, patients were instructed to self-collect saliva into a sterile specimen container, which was then placed on ice until further processing for analysis.

NP swab specimen collections were performed by trained staff at the Hospital of the University of Pennsylvania following the CDC *Interim Guidelines for Collecting and Handling of Clinical Specimens for COVID-19 Testing* (https://www.cdc.gov/coronavirus/2019-nCoV/lab/guidelines-clinical-speci-mens.html- updated as of October 25, 2021).

NP swabs were maintained on ice during the collection period, transferred to cryogenic vials, and stored at –80°C until analysis. RNA was extracted and purified using the QIAmp DSP Viral RNA Mini Kit (Qiagen). The first step of this process inactivated the virus from the NP samples and was performed in a biosafety cabinet under BSL-2 enhanced protocols, while subsequent steps were performed on a lab bench under standard conditions. The RNA was analyzed using the TaqPath 1-Step RT-qPCR reagent (Life Technologies) on the Quantstudio 7 Flex Genetic Analyzer (ABI). The oligonucleotide primers and probes for detection of SARA-CoV-2 (sequences provided in *Supplementary file 1a*) were selected from regions of the virus nucleocapsid (N) gene. The panel is designed for specific detection of the SARA-CoV-2 viral RNA (two primer/probe sets, N1 and N2). An additional primer/probe set to detect the human RNase P gene (RP) in control samples and clinical specimens is also included in the panel (2019-nCoVEUA-01). RNaseP is a single copy human-specific gene and can indicate the amount of human cells collected.

## Paper COV-ID

Squares of Whatman no. 1 filter paper (2 mm × 2 mm) were cut using a scalpel on a clean surface under a laminar flow hood and stored at room temperature until used. Using ethanol-sterilized fine-nosed tweezers a single square was dipped twice into unprocessed, freshly collected saliva with or without added SARS-CoV-2 (BEI Resources Cat. NR-52286) until saliva was saturated on paper by eye. Paper was then transferred to well of 96-well plate containing 10 μL of 1× TCEP/EDTA buffer (2.5 mM TCEP, 1 mM EDTA, 1.15 NaOH). Plate was placed on heat block inside laminar flow hood or inside open thermocycler and incubated at 95°C × 10 min.

Ten microliter RT-LAMP mixture was prepared as described above in the absence of the N2 spike RNA. 10 μL of RT-LAMP reaction mixture was added to each paper strip, then plate was sealed and incubated 65°C × 1 hr, 95°C × 5 min in QuantStudio Flex 7 (Thermo). One microliter of each reaction was either diluted 1:100 or purified via SPRIselect beads and PCR amplified as described above.

## Sequencing

Libraries were sequenced on one of the following Illumina instruments: MiSeq, NextSeq 500, NextSeq 550, NovaSeq 6000 and sequenced using single end programs with a minimum of 40 cycles on Read 1 and 8 cycles for index 1 (on P7) and index 2 (on P5).

## Sequence analysis

Reads were filtered for optical quality using FASTX-toolkit utility fastq_quality_filter (http://hannonlab.cshl.edu/fastx_toolkit/), then cutadapt (*Martin, 2011*) was used to remove adapters and demultiplex LAMP barcodes. Reads were aligned to a custom index containing SARS-CoV-2 genome (NC_045512.2), Influenza H1N1 coding sequences (NC_026431.1, NC_026432.1, NC_026433.1, NC_026434.1, NC_026435.1, NC_026436.1, NC_026437.1, NC_026438.1), STATH coding sequence (NM_003154.3), ACTB coding sequence (NM_001101.5) and custom N2 spike sequence (*Supplementary file 1a*). Alignment was performed using bowtie2 (*Langmead and Salzberg, 2012*) with options `--no-unal` and `--end-to-end`. Alignments with greater than 1 mismatch were removed and the number of reads mapping to each target for all barcodes were extracted and output in a matrix. Barcodes with fewer than 25 total mapped reads were discarded. Alignment script and bowtie2 indexes are provided as *Source code 1*.

## Acknowledgements

The authors thank E Shields for careful proofreading of analysis scripts; B Morris and R Collman for the collection and distribution of clinical saliva samples; F Bushman, S Sherril-Mix, and Abigail Glascock for sharing RT-qPCR data on the clinical samples; the UPenn rapid assay task force for project feedback; the gLAMP weekly forum for advice and guidance; and the Perelman School of Medicine Covid19 Research Fund.

## Additional information

### Competing interests

Benjamin S Abella: Research funding from Becton Dickinson Speaking honoraria from Becton Dickinson, Stryker Equity in VOC Health, a company developing novel COVID19 detection technology distinct from the topic of this manuscript. Roberto Bonasio: Reviewing editor, *eLife*. The other authors declare that no competing interests exist.

### Funding

| Funder | Grant reference number | Author |
|---|---|---|
| National Heart, Lung, and Blood Institute | HL147123 | Parisha P Shah<br>Rajan Jain |
| American Heart Association | | Parisha P Shah<br>Rajan Jain |
| Burroughs Wellcome Fund | | Parisha P Shah<br>Rajan Jain |
| Allen Foundation | | Parisha P Shah<br>Rajan Jain |

This study was made possible by an internal Perelman School of Medicine Covid19 Research Fund. The funders had no role in study design, data collection and interpretation, or the decision to submit the work for publication.

### Author contributions

Robert Warneford-Thomson, Conceptualization, Formal analysis, Investigation, Methodology, Software, Validation, Visualization, Writing – original draft, Writing – review and editing; Parisha P Shah, Investigation, Methodology, Validation, Writing – original draft, Writing – review and editing; Patrick Lundgren, Investigation, Validation, Writing – review and editing; Jonathan Lerner, Investigation,

Writing – review and editing; Jason Morgan, Antonio Davila, Investigation; Benjamin S Abella, Resources, Writing – review and editing; Kenneth Zaret, Rajan Jain, Supervision, Writing – review and editing; Jonathan Schug, Methodology, Software, Supervision, Writing – review and editing; Christoph A Thaiss, Roberto Bonasio, Conceptualization, Project administration, Supervision, Writing – original draft, Writing – review and editing

## Author ORCIDs
Robert Warneford-Thomson (iD) http://orcid.org/0000-0002-4521-0568
Parisha P Shah (iD) http://orcid.org/0000-0002-7756-6868
Patrick Lundgren (iD) http://orcid.org/0000-0003-1014-8941
Rajan Jain (iD) http://orcid.org/0000-0002-1979-044X
Roberto Bonasio (iD) http://orcid.org/0000-0002-0767-0889

## Ethics
Human subjects: Clinical saliva samples were obtained and characterized as part of a separate study at the University of Pennsylvania and collected under Institutional Review Board (IRB)-approved protocols (IRB protocol #842613 and #813913).

## Decision letter and Author response
Decision letter https://doi.org/10.7554/eLife.69949.sa1
Author response https://doi.org/10.7554/eLife.69949.sa2

---

## Additional files

### Supplementary files
- Supplementary file 1. Oligonucleotide sequences and RT-LAMP indexes.
- Transparent reporting form
- Source code 1. Scripts and Bowtie indexes.

### Data availability
Next generation sequencing data generated for this study are available at the NCBI GEO with accession GSE172118.

The following dataset was generated:

| Author(s) | Year | Dataset title | Dataset URL | Database and Identifier |
|---|---|---|---|---|
| Bonasio R, Warneford-Thomson R | 2021 | COV-ID: A LAMP sequencing approach for high-throughput co-detection of SARS-CoV-2 and influenza virus in human saliva | https://www.ncbi.nlm.nih.gov/geo/query/acc.cgi?acc=GSE172118 | NCBI Gene Expression Omnibus, GSE172118 |

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
