## [Editor Report]

Surveillance screening can help us estimate the prevalence of SARS-CoV-2 infection and co-infection with other respiratory pathogens. This work offers a high-throughput and cost-effective method to do such surveillance based on RT-LAMP combined with deep sequencing. This method can be applied to clinical samples for an accurate reading of the fraction of infections where the SARS-CoV-2 titer is moderate or high.

---

## [Decision Letter]

**Decision letter after peer review:**

Thank you for submitting your article "COV-ID: A LAMP sequencing approach for high-throughput co-detection of SARS-CoV-2 and influenza virus in human saliva" for consideration by *eLife*. Your article has been reviewed by 2 peer reviewers, one of whom is a member of our Board of Reviewing Editors, and the evaluation has been overseen by Dominique Soldati-Favre as the Senior Editor. The reviewers have opted to remain anonymous.

Essential revisions:

The reviewers assessment was that, while the approach may make an important contribution to genomic surveillance of SARS-CoV-2, it was unclear if it would deliver the benefits of decreased cost, time, and labor in a real-world situation.

The main weakness pointed out by both reviewers was that much of the work was done on spiked samples. High throughput on clinical samples was not demonstrated, as only 8 were used.

The reviewers agree that the type of clinical sample tested, whether saliva or swab, is less important. What the proof of concept would require is at least a "superpool" of several pooled wells, with each well containing the maximum number of samples (or close to it). This way, both pooling steps are tested.

*Reviewer #1 (Recommendations for the authors):*

I do not have specific technical concerns, but the methodology is difficult to evaluate with results from only 8 clinical samples. Factors such as differing Ct values, sample quality, and others should be evaluated to see if the approach could be used with high throughput, and this would require at a minimum a "superpool" of several pooled wells with each well containing the maximum number of samples. Participant Ct values should be presented and be correlated to detection. Swab transport media can substitute if saliva is not available. Negative samples should be combined with qPCR confirmed positive samples and false negative and false positive rates determined.

Given that a fast turnaround time is key to success, the approach proposed by the authors is unlikely to be used except in very large studies with ready access to deep sequencing, in which case they would also have access to a PCR. Where the approach may fill an unmet need is genomic surveillance of the virus in the population which would likely involve reconfiguring the system to sequence spike.

*Reviewer #2 (Recommendations for the authors):*

I enjoyed reading this manuscript and was impressed with its potential to make a significant contribution to population-level testing of infectious disease, e.g. during the COVID-19 pandemic. Assuming that you are able to able to address the points in the Public Review, I would be pleased to recommend this manuscript for publication in *eLife*. In particular, I think it would be very helpful to perform more testing of your method from clinical samples. I realise that these may be difficult to acquire and so I do not want to be too demanding in the exact source of these samples, just something a bit more realistic than the spike-ins used extensively here.

[Editors' note: further revisions were suggested prior to acceptance, as described below.]

Thank you for resubmitting your article "A LAMP sequencing approach for high-throughput co-detection of SARS-CoV-2 and influenza virus in human saliva" for consideration by *eLife*. Your revised article has been reviewed by 2 peer reviewers, one of whom is a member of our Board of Reviewing Editors, and the evaluation has been overseen by Dominique Soldati-Favre as the Senior Editor. The reviewers have opted to remain anonymous.

The main comment of the Reviewers was to show feasibility with more than 8 clinical samples, given that the reason for the approach was to do high-throughput analysis of clinical samples.

The authors therefore used 120 clinical samples in an experiment to show that the approach can be scaled up. The results are presented in Figure 2—figure supplement 2.

While the reviewer comments were addressed, the results seem to show that the sensitivity of the approach (rate of false negatives) is considerably inferior to that of qPCR: By counting on panel A of the figure, there are 17 qPCR-positive samples. Thresholding by the highest values of SARS/Spike+1 in the qPCR negative samples (about 0.1), 8 samples out of 17 are above threshold by the authors' approach. Therefore, 9 out of 17 (53%) are false negative.

Because the sample is imbalanced – many more qPCR negatives relative to positives – the false-positive rate (1-specificity) is not sensitive to false positives and the ROC curves are overly optimistic in describing the result.

The reviewers agreed that those intending to implement the approach can decide for themselves whether it is right for them, but that these limitations should be clearly stated. The authors can use a Precision-Recall curve which does not consider the false positive rate to quantify the performance of their approach.

Also, the strategy of an artificial N2 spike-in is not described until later in the MS. It should be described for the figure to explain the SARS/Spike+1 ratio.

*Reviewer #1 (Recommendations for the authors):*

The main comment of the Reviewers was to show feasibility with more than 8 clinical samples, given that the reason for the approach was to do high-throughput analysis of clinical samples.

The authors therefore used 120 clinical samples in an experiment to show that the approach can be scaled up. The results are presented in Figure 2—figure supplement 2.

Using qPCR as the gold standard, there are 17 samples (estimated, from counting the points) out of 120 (14%) that had a detectable Ct value. Using the presented approach without thresholding, there were 86 samples out of 120 (72%) where viral sequences were detected.

Thresholding by the highest values of SARS/Spike+1 in the qPCR negative samples (about 0.1), 8 samples out of 120 (7%) were above threshold, with 9 out of 17 (53%) being false negative.

If this is a misinterpretation of the results, then I would suggest clarifying it. If not, the method presented has low sensitivity, unless indeed the majority of people in the cohort had SARS-CoV-2 infections not detected by qPCR.

Also, the strategy of an artificial N2 spike-in is not described until later in the MS. It should be described for the figure to explain the SARS/Spike+1 ratio. The name of the denominator should perhaps be changed to avoid confusion with the SARS-CoV-2 spike gene. The horizontal line at 0.2 which appears to be a threshold in Figure 2—figure supplement 2B should be explained and the scale changed to log to make all the values visible and not have a scale interruption.

Reviewer #2 (Recommendations for the authors):

I thank the authors for their efforts in revising this manuscript and particularly for performing another round of sample collection/analysis. The results presented in Figure 2 - Figure Supplement Figure 2 significantly improve the statistical confidence in the ability of this method to identify positive clinical samples and the ROC plots will allow readers to make their own quantitative judgements of the performance of this method for their own purposes. I also thank the authors for making all the minor edits I suggested in my previous review.

I now fully support publication of this article in *eLife* as is.

---

## [Author Response]

Essential revisions:The reviewers assessment was that, while the approach may make an important contribution to genomic surveillance of SARS-CoV-2, it was unclear if it would deliver the benefits of decreased cost, time, and labor in a real-world situation.The main weakness pointed out by both reviewers was that much of the work was done on spiked samples. High throughput on clinical samples was not demonstrated, as only 8 were used.The reviewers agree that the type of clinical sample tested, whether saliva or swab, is less important. What the proof of concept would require is at least a "superpool" of several pooled wells, with each well containing the maximum number of samples (or close to it). This way, both pooling steps are tested.

We have now performed COV-ID on 120 additional clinical samples that were collected at the Hospital of the University of Pennsylvania (new Figure 2—figure supplement 2). Because the saliva collection project was terminated, we could not obtain additional clinically validated samples of saliva; therefore, these samples were collected in the more common form of nasopharyngeal (NP) swabs.

To test the “superpool” strategy, we used up to 10 unique LAMP barcodes in 13 PCR pools, which were subsequently combined (i.e. “superpooled”) in a single sequencing library. The results showed a strong correlation of COV-ID read ratios with state-of-the-art individual qPCR quantification of viral RNA (new Figure 2—figure supplement 2A), with very good sensitivity and specificity (new Figure 2—figure supplement 2C).

Reviewer #1 (Recommendations for the authors):I do not have specific technical concerns, but the methodology is difficult to evaluate with results from only 8 clinical samples. Factors such as differing Ct values, sample quality, and others should be evaluated to see if the approach could be used with high throughput, and this would require at a minimum a "superpool" of several pooled wells with each well containing the maximum number of samples. Participant Ct values should be presented and be correlated to detection. Swab transport media can substitute if saliva is not available. Negative samples should be combined with qPCR confirmed positive samples and false negative and false positive rates determined.

We have now added results from 120 additional clinical samples from NP swabs (new Figure 2-figure supplement 2). For each of 13 pooled PCRs, we combined RT-lamps from up to 10 patient samples. Although the experimenter was blinded at the time of pooling, in retrospect we determined that of the 13 pools 5 contained at least one qPCR-confirmed positive sample mixed with negative samples as shown in the new figure (Figure 2—figure supplement 2B). False negative and false positive rates depend on the Ct threshold chosen for the clinical qPCR to define “true positive”. ROC plots are shown for three potential thresholds (new Figure 2—figure supplement 2C).

Given that a fast turnaround time is key to success, the approach proposed by the authors is unlikely to be used except in very large studies with ready access to deep sequencing, in which case they would also have access to a PCR.

To analyze thousands or tens of thousands of individual samples per day by qPCR, a lab would need to have access not just to one PCR but to many qPCR instruments. Because the first step of COV-ID, which introduces patient-level barcodes by RT-LAMP, does not require a PCR and can be performed separately, we envision the possibility that the first step could be performed on site or in a minimally equipped satellite lab and the samples could be pooled and sequenced in a central location. This would naturally require more development and logistics, but we think it is an inherent advantage of COV-ID (and to some extent of LAMP-seq).

We have clarified and discussed these views in the revised discussion (lines 372–4).

Where the approach may fill an unmet need is genomic surveillance of the virus in the population which would likely involve reconfiguring the system to sequence spike.

We agree and we mention this possibility in the discussion (lines 345–50).

Reviewer #2 (Recommendations for the authors):I enjoyed reading this manuscript and was impressed with its potential to make a significant contribution to population-level testing of infectious disease, e.g. during the COVID-19 pandemic. Assuming that you are able to able to address the points in the Public Review, I would be pleased to recommend this manuscript for publication in eLife. In particular, I think it would be very helpful to perform more testing of your method from clinical samples. I realise that these may be difficult to acquire and so I do not want to be too demanding in the exact source of these samples, just something a bit more realistic than the spike-ins used extensively here.

Indeed, the saliva sample collection at Penn ended before we could obtain more samples. we therefore tested COV-ID on 120 NP-derived samples (new Figure 2—figure supplement 2).

[Editors' note: further revisions were suggested prior to acceptance, as described below.]

The reviewers have discussed their reviews with one another, and the Reviewing Editor has drafted this to help you prepare a revised submission.The main comment of the Reviewers was to show feasibility with more than 8 clinical samples, given that the reason for the approach was to do high-throughput analysis of clinical samples.The authors therefore used 120 clinical samples in an experiment to show that the approach can be scaled up. The results are presented in Figure 2—figure supplement 2.While the reviewer comments were addressed, the results seem to show that the sensitivity of the approach (rate of false negatives) is considerably inferior to that of qPCR: By counting on panel A of the figure, there are 17 qPCR-positive samples. Thresholding by the highest values of SARS/Spike+1 in the qPCR negative samples (about 0.1), 8 samples out of 17 are above threshold by the authors' approach. Therefore, 9 out of 17 (53%) are false negative.

We agree that, on samples from NP swabs (COV-ID was originally developed for saliva), our method is not as sensitive as individual qPCRs. We now state this in the text (lines 234–5). However, when considering false negatives, we note that 7 of them have Ct values > 36 by qPCR and 9 have Ct values > 31. Other groups who have published sequencing-based high-throughput COVID testing approaches, such as SARSeq and LAMP-seq (Ludwig et al., 2021; Yelagandula et al., 2021), have reported difficulties in scoring samples with high qPCR Ct values. The SARSeq authors specifically show that Ct’s > 36 are not even reproducible by TaqMan qPCR (Figure 4E in Yelagandula et al., 2021). We have added these considerations to the text (lines 236–8) and colored the circles to reflect the different Ct thresholds (Figure 2—figure supplement 3A–B).

Because the sample is imbalanced – many more qPCR negatives relative to positives – the false-positive rate (1-specificity) is not sensitive to false positives and the ROC curves are overly optimistic in describing the result.The reviewers agreed that those intending to implement the approach can decide for themselves whether it is right for them, but that these limitations should be clearly stated. The authors can use a Precision-Recall curve which does not consider the false positive rate to quantify the performance of their approach.

We have included precision-recall curves (new Figure 2—figure supplement 3D) and refer to them in the revised text (line 240).

Also, the strategy of an artificial N2 spike-in is not described until later in the MS. It should be described for the figure to explain the SARS/Spike+1 ratio.

We agree. We moved the description of the N2 spike-in (now referred to as “synthetic calibration standard” based on Reviewer 1’s suggestion, see below) earlier in the manuscript prior to discussion of the clinical NP swab samples (lines 194–225).

Reviewer #1 (Recommendations for the authors):The main comment of the Reviewers was to show feasibility with more than 8 clinical samples, given that the reason for the approach was to do high-throughput analysis of clinical samples.The authors therefore used 120 clinical samples in an experiment to show that the approach can be scaled up. The results are presented in Figure 2—figure supplement 2.Using qPCR as the gold standard, there are 17 samples (estimated, from counting the points) out of 120 (14%) that had a detectable Ct value. Using the presented approach without thresholding, there were 86 samples out of 120 (72%) where viral sequences were detected.Thresholding by the highest values of SARS/Spike+1 in the qPCR negative samples (about 0.1), 8 samples out of 120 (7%) were above threshold, with 9 out of 17 (53%) being false negative.If this is a misinterpretation of the results, then I would suggest clarifying it. If not, the method presented has low sensitivity, unless indeed the majority of people in the cohort had SARS-CoV-2 infections not detected by qPCR.

We agree that, on samples from NP swabs (COV-ID was originally developed for saliva), our method is not as sensitive as individual qPCRs. We now state this in the text (lines 234–5). However, when considering false negatives, we note that 7 of them have Ct values > 36 by qPCR and 9 have Ct values > 31. Other groups who have published sequencing-based high-throughput COVID testing approaches, such as SARSeq and LAMP-seq (Ludwig et al., 2021; Yelagandula et al., 2021), have reported difficulties in scoring samples with high qPCR Ct values. The SARSeq authors specifically show that Ct’s > 36 are not even reproducible by TaqMan qPCR (Figure 4E in Yelagandula et al., 2021). We have added these considerations to the text (lines 236–8) and colored the circles to reflect the different Ct thresholds (Figure 2—figure supplement 3A–B).

Also, the strategy of an artificial N2 spike-in is not described until later in the MS. It should be described for the figure to explain the SARS/Spike+1 ratio. The name of the denominator should perhaps be changed to avoid confusion with the SARS-CoV-2 spike gene.

We agree. We moved the description of the N2 spike-in earlier in the manuscript prior to discussion of the clinical NP swab samples (lines 194–225). We also agree that referring to this as spike-in might confuse readers and we changed it to “synthetic calibration standard”.

The horizontal line at 0.2 which appears to be a threshold in Figure 2—figure supplement 2B should be explained and the scale changed to log to make all the values visible and not have a scale interruption.

We have converted the *y* axis to a log scale (Figure 2—figure supplement 3B). We now explain in the legend that the line represents a signal threshold 10-fold higher than the highest negative control. A more accurate view of sensitivity and specificity is provided by the ROC and precision recall curves, which do not rely on thresholds (Figure 2—figure supplement 3C-D).